# C24 Ceramide Lipid Nanoparticles for Skin Wound Healing

**DOI:** 10.3390/pharmaceutics17020242

**Published:** 2025-02-12

**Authors:** Ji-Hye Lee, Jin-Hyun Kim, Tong-Il Hyeon, Khee-Tae Min, Se-Young Lee, Han-Chul Ko, Hong-Seok Choi, Kuk-Youn Ju, Young-Seok Cho, Tae-Jong Yoon

**Affiliations:** 1Moogene Medi Institute, 25, Misagangbyeonjungang-ro 7beonan-gil, Hanam 12939, Republic of Korea; jhlee@moogene.com (J.-H.L.); samkim@moogene.com (J.-H.K.); hti327@moogene.com (T.-I.H.); kt0807@moogene.com (K.-T.M.); sylee@moogene.com (S.-Y.L.); hanchul0713@moogene.com (H.-C.K.); hschoi@moogene.com (H.-S.C.); jky00@moogene.com (K.-Y.J.); 2Division of Gastroenterology, Department of Internal Medicine, Seoul St. Mary’s Hospital, College of Medicine, The Catholic University of Korea, 505 Banpo-dong, Seoul 06591, Republic of Korea; 3Research Institute of Pharmaceutical Science and Technology (RIPST), College of Pharmacy, Ajou University, 206 Worldcup-ro, Yeongtong-gu, Suwon 16499, Republic of Korea; 4Department of BioHealth Regulatory Science, Graduate School of Ajou University, 206 Worldcup-ro, Yeongtong-gu, Suwon 16499, Republic of Korea

**Keywords:** C24 ceramide, lipid nanoparticle, re-epithelialization, wound healing, collagen regeneration

## Abstract

**Background/Objectives**: C24 ceramide plays a crucial role in skin regeneration and wound healing; however, its hydrophobic nature limits its application in therapeutic formulations. This study aims to enhance the bioavailability and efficacy of C24 ceramide by developing ceramide-based lipid nanoparticles (C24-LNP) and evaluate their impact on skin regeneration and wound healing. **Methods**: C24-LNP was synthesized and characterized for aqueous stability and bioavailability. In vitro experiments were conducted to assess its effects on keratinocyte proliferation and migration. Molecular biological analysis examined key signaling pathways, including AKT and ERK1/2 phosphorylation. Additionally, an in vivo mouse wound model was utilized to evaluate wound healing efficacy, with histological analysis performed to assess epidermal and dermal regeneration. **Results**: C24-LNP exhibited improved aqueous stability and bioavailability compared to free C24 ceramide. In vitro studies demonstrated that C24-LNP significantly promoted keratinocyte proliferation and migration. Molecular analysis revealed activation of the AKT and ERK1/2 signaling pathways, which are critical for cell growth and skin regeneration. In vivo wound healing experiments showed that C24-LNP accelerated wound closure compared to the control group. Histological analysis confirmed enhanced epidermal and dermal regeneration, leading to improved structural and functional skin repair. **Conclusion**: The lipid nanoparticle formulation of C24 ceramide effectively increases its bioavailability and enhances its therapeutic efficacy in skin regeneration and wound healing. C24-LNP presents a scalable and cost-effective alternative to traditional growth factor-based therapies, offering significant potential for clinical applications in wound care and dermatological treatments.

## 1. Introduction

The total surface area of the skin of an adult is approximately 1.5–2 m^2^, making it the largest organ in the body in terms of surface area. The skin is composed of the epidermis, dermis, and subcutaneous fat. It maintains the body’s moisture and temperature and protects the internal from the external environment, such as chemicals, heat, ultraviolet rays, and microorganisms [1]. When a wound, like an abrasion, occurs, the skin loses its protective function, allowing for potential internal infections, so it must heal quickly. Skin wound healing proceeds through four major stages: hemostasis, inflammation, proliferation, and remodeling [2]. Each stage involves a series of cellular and biochemical reactions that repair damaged tissue and regenerate new skin. Each stage is essential for effective healing, allowing the skin to regain its original shape and function over time. Given the need for this rapid healing, interest in ceramides has recently increased. Ceramides are essential chemicals that maintain the skin’s protective barrier. Ceramides are mainly found in the outermost layer of the skin (stratum corneum) and make up approximately 50% of the lipids that fill the spaces between skin cells [3]. Ceramides perform various functions, such as moisture retention, barrier strengthening, and skin elasticity [4,5]. For example, when the skin becomes dry or the barrier is damaged, its ceramide level decreases, making the skin more susceptible to irritation and inflammation. Therefore, supplying ceramides to the skin is crucial [6].

Ceramides have a unique sphingoid backbone bound to fatty acid chains [7,8]. The length and saturation of these fatty acid chains affect the chemical properties of ceramides, such as its fluidity and moisturizing properties. For example, ceramides with long-alkyl-chain fatty acids can enhance the barrier strength, while short-chain ceramides can improve skin elasticity [9]. Ceramides with long fatty acids are known to have unique functions. Among them, C24 ceramide (C24) has 24 alkyl chains in the fatty acid moiety and is chemically stable and non-irritating, making it suitable for various skin types. In addition, C24 is particularly useful for repairing and strengthening damaged skin barriers [10]. It helps to quickly restore barrier function, reduce skin irritation, and improve overall health. However, due to its relatively long alkyl fatty acid chain, C24 is poorly soluble in aqueous solutions. Therefore, C24 ceramide must be applied using organic solvents such as alcohol, which can dry out the skin and make its effective application challenging. This study determines whether C24 has additional cell growth and migration capabilities beyond its previously known functions.

Recently, many studies have been conducted to improve the physical properties of C24, such as the solubility of its raw materials through nanoparticulation [11]. Ceramide is manufactured into lipid nanoparticles such as liposomes to effectively provide hydrophilic particles. In addition, the properties of nanoparticles can be expected to effectively penetrate the skin or cells to provide the benefits of ceramide [12,13,14,15,16]. Previous studies have explored various formulations to enhance ceramide delivery, including microemulsion, nanoemulsion, and the use of liposomes and nanoparticles. While these approaches have improved solubility and stability, challenges such as poor skin penetration, instability, and the need for high surfactant concentrations remain. In particular, C24 ceramide, despite its potent barrier-repairing properties, has been difficult to apply due to its hydrophobic nature. Reports have also shown that ceramides in the form of hydrophobic substances may be detrimental to tissue regeneration [3]. In this report, hydrophobic C24 ceramide was formulated with LNPs for water-soluble C24 ceramide, and the synthesized C24-LNPs showed high bioavailability and exhibited effective wound-healing functions in vitro and in vivo. The ceramide delivery system of the proposed method can transport ceramides in aqueous media such as micelles, microemulsions, and other delivery vesicles.

## 2. Materials and Methods

### 2.1. Preparation and Characterization of C24-LNP

Ceramide-based lipid nanoparticles were prepared via ethanol injection methods [17]. Briefly, distearoylphosphatidylcholine (DSPC, Croda, Seoul, Republic of Korea, 8.61 mg), cholesterol (Samchun Chemicals, Seoul, Republic of Korea, 12.31 mg), N-lignoceroyl-D-erythro-sphingosine (C24 ceramide, Avanti Polar Lipids, Alabaster, AL, USA, 23.96 mg), and 1,2-dipalmitoyl-sn-glycero-3-phospho-(1′-rac-glycerol) sodium salt (DPPG-Na, NOF, 3.12 mg) were dissolved in ethanol solution (3 mL) with a molar ratio of 13:38:44:5 and then added gently to deionized water (9 mL) with a volume ratio of 3:1. After agitation with continuous stirring at 500 rpm using a magnetic stirrer for 10 min, the mixture solution (12 mL) was treated to remove the ethanol via rotary evaporation (60 ± 5 °C). The production of the bare LNPs used a process almost similar to that used for producing C24-LNPs; the composition ratios are shown in Figure 1A. The final LNP solution (10 mg/mL) was characterized based on its particle size, polydispersity index, and zeta potential using a Zeta-sizer Nano ZS (Malvern, Seoul, Republic of Korea). Cryo-EM images of the C24-LNPs for microscopy were prepared by placing 5 μL of LNPs onto a 400-mesh lacey carbon-coated grid using single-sided blotting for 2 s; then, the sample grid was immediately immersed into nitrogen-cooled ethane. The C24-LNP morphology was then observed using a Krios G4 (Thermo Fisher Scientific, Seoul, Republic of Korea) at liquid nitrogen temperature and 300 kV.

### 2.2. Cell Culture

Immortalized human skin keratinocytes, HaCaT purchased from the Korean Cell Line Bank (Seoul, Republic of Korea), and the cell were cultured in Dulbecco’s modified eagle medium (DMEM) supplemented with 10% fetal bovine serum (FBS) and 1% penicillin–streptomycin (PS) at 37 °C under 5% CO_2_. The cells were sub-cultured at 70–80% confluency and used with a passage below 30. The cells were tested negative for mycoplasma.

### 2.3. Cell Proliferation Assay

The cells were seeded in a 96-well cell culture plate (3 × 10^4^ cells) at 37 °C under 5% CO_2_. The cells were treated with various concentrations of C24-LNP for 24 h at 37 °C, and cell viability was measured with a WST-8 cell counting kit (Dojindo, Kumamoto, Japan). The absorbance was measured at 450 nm using a microplate reader (BioTek, Seoul, Republic of Korea), and the negative control group’s relative survival rate (viability %) was calculated.

### 2.4. Cell Migration Assay

To assess the wound-healing effects of C24-LNP in vitro, HaCaT human skin keratinocyte cells were seeded (24-well culture plate, SPL) and cultured in reduced serum for 24 h. Then, the cells were scratched using a scratcher (Scar Scratcher, SPL). After 24 h, the cells were treated with different concentrations of C24-LNP for 48 h. The wound area was measured on days 0, 1, and 2 after the scratch. The human recombinant EGF protein (Thermo Fisher Scientific, Seoul, Republic of Korea, 10 ng/mL), bare LNP without C24 ceramide (100 μg/mL), or each LNP component (100 μg/mL) was treated via scratching and used as positive and negative controls. The recovered wound area was calculated using Image-J Software (Version 1.54m). The data were statistically evaluated from three independent experiments in three replicates.

### 2.5. Western Blot

The cells were treated with C24-LNP (500 µg/mL) for 24 h and then lysed in a RIPA buffer containing a protease inhibitor cocktail (Sigma-Aldrich, Seoul, Republic of Korea). A BCA assay kit was used to measure the total protein amounts (Thermo Fisher Scientific, Seoul, Republic of Korea). Samples containing equal amounts of protein were separated using SDS-PAGE on a 4–15% gradient gel (Bio-Rad, XT-Criterion, Seoul, Republic of Korea) and transferred onto polyvinylidene difluoride (PVDF) membranes. After the transfer, the membranes were blocked with 5% skim milk in TBS-T (Tris buffer saline-0.02% Tween) buffer for 1 h at room temperature. The following primary antibodies were used to determine the proliferation and their signal pathway: anti-Collagen I (Cell Signaling Technology, Danvers, MA, USA, rabbit-monoclonal, 1:1000), anti-Cytokeratin-14 (Santa Cruz Biotechnology, Heidelberg, Germany, mouse-monoclonal, 1:200), anti-Ki-67 (Santa Cruz Biotechnology, Heidelberg, Germany, mouse-monoclonal, 1:200), anti-PCNA (Santa Cruz, mouse-monoclonal, 1:200), anti-AKT (Cell Signaling Technology, Danvers, MA, USA, rabbit-polyclonal, 1:1000), anti-pAKT (Cell Signaling Technology, Danvers, MA, USA, rabbit-polyclonal, 1:1000), anti-ERK1/2 (Cell Signaling Technology, Danvers, MA, USA, rabbit-polyclonal, 1:1000), and anti-pERK1/2 (Cell Signaling Technology, Danvers, MA, USA, mono-polyclonal, 1:1000). Anti-β-actin (Santa Cruz Biotechnology, Heidelberg Germany, mouse-monoclonal, 1:10,000) was used as an internal loading control. Each primary antibody was incubated overnight at 4 °C. Following three washes with TBS-T buffer, a reaction with appropriate horseradish peroxidase (HRP)-conjugated mouse or rabbit secondary antibodies (Santa Cruz Biotechnology, Heidelberg, Germany) was carried out for 2 h at room temperature and finally detected using a Pierce ECL kit (Middleton, MA, USA). Bands were imaged via chemiluminescence imaging using an ImageQuant LAS500 (GE Healthcare, Chicago, IL, USA).

### 2.6. Animals

Male HR-1 hairless mice aged six weeks and weighing 18–20 g were purchased from Central Laboratory Animal Inc. (Seoul, Republic of Korea). All procedures followed the local institutional guidelines for animal care established by the Catholic University Animal Use and Care Committee (IACUC NO: 2024-0209-01). The mice were allowed to adapt to the environment for 1 week before the experiment. They were housed in cages with controlled temperatures (24–25 °C), normal light–dark cycles, and 60% humidity. In an in vivo animal model, full-thickness nude mouse skin wounds were induced on the back of the mice using a 6 mm biopsy punch (Mckesson, Irving, TX, USA) under isoflurane anesthesia (Hana Pharma Co., Ltd., Seoul, Republic of Korea) to determine the extent of wound healing. The mice were randomly assigned to the following four groups: non-treated negative control, EGF-protein-treated positive control, C24-LNP, and bare LNP without C24 ceramide.

### 2.7. In Vivo Efficacy Assay

To determine the therapeutic potential of C24-LNP for skin wounds, 10 µL of C24-LNP (1 mg/mL in DW, 4.4 µg C24) was applied topically daily for 10 days starting from the day of injury (day 0) according to previous in vivo wound-healing experiments [18]. The wound was covered with a transparent medical dressing (Tegaderm film, 3M). The wound size, as seen above, was measured on days 2, 4, 6, 8, and 10 after injury, and the recovery rate (%) was evaluated based on the degree of contraction of the wound area through photographic images measured using Fiji Image-J. Regeneration of the epithelium and dermis was assessed by histology and Western blot on the 4th and 10th days post-injury, respectively, compared with the control group. A Western blot analysis was performed using skin tissue lysates using the same procedure as a cell lysate analysis.

### 2.8. Hematoxylin and Eosin (H&E) Staining

The excised tissues were fixed with 4% neutral paraformaldehyde and embedded in paraffin. The sections were cut at 6 µm thickness and deparaffinized according to the general protocol. The stained slides were observed using a Nikon Eclipse E200 microscope (Nikon, Tokyo, Japan).

### 2.9. Masson’s Trichrome (MT) Collagen Staining

MT staining was performed using paraffin sections to evaluate the collagen production effect of C24-LNP on skin dermal regeneration during wound healing. The protocol and histology analysis are similar to those of H&E. The amount of blue-stained collagen in the dermal layer was quantified using ImageJ software (Version 1.54m).

### 2.10. Statistical Analysis

All values are expressed as mean ± SEM (standard error of the mean). The statistical analyses were performed using Graph Pad Prism9 (GraphPad Software Inc., La Jolla, CA, USA). In vitro cell proliferation, migration, and in vivo wound-healing rate (%) were assessed via an analysis of variance (ANOVA), followed by Tukey’s or Dunnett’s multiple comparison test with a significance level (α) of 0.05.

## 3. Results

### 3.1. Preparation and Characterization of C24-LNP

We synthesized C24-LNP with appropriate ratios of the neutral lipid, cholesterol, and anionic lipid compound (Figure 1A) [19]. To manufacture LNPs with high biocompatibility, we tried to exclude the use of cationic lipid compounds known to have cytotoxicity as much as possible. In addition, we attempted to maintain the cholesterol content as much as possible to maintain structural stability. We confirmed that a minimum of 5% of anionic lipid compounds, such as DPPG, must have aqueous solution stability. Finally, we have established the composition of the optimized C24-LNP. The nanoparticles composed of the optimal ratio exhibited very high aqueous stability. The analytical results showed that the average uniform size was 127 ± 5 nm, and their poly-diversity index (PDI) value was determined at 0.165 ± 0.007 via dynamic light scattering (DLS) analysis. The surface charge of the synthesized C24-LNP showed an anionic character of −51 ± 2 mV. From the cryo-EM analysis, we confirmed that the C24-LNPs were spherical, had a relatively uniform size distribution, and exhibited a bilayer structure (Figure 1B).

### 3.2. Proliferation and Migration Effect of C24-LNP on Human Keratinocytes

HaCat cells are immortalized human keratinocytes and have been extensively used to study epidermal homeostasis and its pathophysiology. Human keratinocytes treated with C24-LNP were subjected to in vitro assays. Cell viability was examined according to the treatment concentration of C24-LNP, and the cell growth rate increased rapidly compared with the control at concentrations above 50 µg/mL (Figure 2A). Cell growth reached saturation at 500 µg/mL, indicating that the effect depended on the treatment concentration of C24-LNP. We examined C24-LNP uptake into keratinocytes through energy-dependent endocytosis and whether the released C24 ceramide in the cytoplasm affected signaling related to cell proliferation (Appendix A). In addition, we confirmed the degree of effect by comparing it with the treatment using EGF protein, which is known to affect cell growth (Figure 2B and Appendix A). When we treated the LNP, which consists of the absence of only the C24 ceramide component (called bare LNP), the effect on cell growth and biomarkers was small. These results suggest that C24 ceramide leakage into the cytoplasm affects cell growth. According to the Western blot analysis, the expression of biomarkers related to proliferation (Ki-67, PCNA, AKT, and ERK1/2) increased in the treatment group, and the expression levels were similar to or relatively higher than those of the positive group treated with epidermal growth factor (EGF) protein, which is known to promote cell growth. We observed that LNPs containing C24 ceramide can sufficiently increase the proliferation of cell lines in vitro without the biological drug EGF protein. EGF protein generally binds to EGFR in the cell membrane, affects RAS, and transmits signals to AKT and ERK to promote cell growth.

On the other hand, C24 ceramide directly affects ERK and AKT expression in the cytoplasm, regardless of the receptor on the cell membrane. We also investigated whether migration increases with cell growth stimulation. This experiment may indicate the extent to which wounds heal in the actual epidermis. Therefore, in vitro migration assays may indicate the effectiveness of wound-dressing therapies [20,21]. We investigated the extent of recovery via migration after culture by removing cells at 1500 µm intervals using a scratcher (Figure 2C and Appendix A). The extent of recovery varied depending on the treatment concentration of C24-LNP, and a significant migration effect was observed compared with the control at concentrations above 500 µg/mL. In particular, the high-concentration C24-LNP treatment group of 1 mg/mL showed a recovery effect similar to that of the EGF protein drug treatment group. However, when each component of the non-LNP-formulated C24-LNP was treated in the cells, the migration effect was not enhanced (Appendix A). Because the solubility of C24 ceramide is inherently low, the cellular uptake efficiency at the molecular level is considerably lower than that of the nanoparticle formulation. Furthermore, when treated with bare LNP without only C24 ceramide, a migration effect similar to that of the negative control was observed. These results demonstrate that nanoparticles containing C24 ceramide enhance both cell growth and migration effects related to cellular uptake.

### 3.3. In Vivo Wound Healing Effects

The wound-healing effect of C24-LNP was evaluated using an animal model with a 6 mm biopsy punch-induced wound. The treatment groups received 10 μL of either C24-LNP or bare LNP (1 mg/mL). As a positive control, EGF (100 μg/mL) was applied at the same volume (10 μL). Wound recovery was monitored over time (Figure 3). On the fourth day, the C24-LNP and EGF groups showed significantly faster healing effects than the other groups. On the 10th day, while all treatment groups exhibited some level of recovery, the C24-LNP group achieved the highest recovery rate (86.9 ± 2.2%) compared with the control group (75 ± 1.2%), the EGF group (80.3 ± 0.4%), and the bare LNP group (74.2 ± 2.0%). (Mean ± SEM, *n* = 3, ** *p* < 0.05 vs. control or bare LNP.) This is thought to be observed because the dermal layer growth process is dominant until the fourth day, when the epidermal layer grows rapidly thereafter.

### 3.4. Histological Analysis

The healing effect on the skin was evaluated using a histological analysis of the treatment groups. An analysis of the H&E-stained sections confirmed that C24-LNP accelerated re-epithelialization on day 4 post-injury (Figure 4A,B). The epithelium of the C24-LNP and EGF-positive treatment groups recovered faster and more effectively to typical skin structure than the control group. In particular, the epithelium in the C24-LNP group showed significant thickness (37.0 ± 6.3 µm), and the dermal structure recovered rapidly. Keratinocytes and fibroblasts increased in the epidermis and dermis, which dominated cell flow or migration in the normal tissue area and induced proliferation.

In contrast, the control and bare LNP-treated groups showed slower recovery and were characterized by a thinner epithelial layer and an unclear dermal structure. Changes in the proliferation-related biomarkers in the tissue were confirmed by a Western blot analysis in an in vivo animal model (Figure 4C and Appendix A). Similar to the in vitro results, the expression of most proliferation-related markers increased, with higher expression observed in the C24-LNP-treated group than in the EGF-treated group. The increase in proliferating cell nuclear antigen (PCNA), AKT, and ERK1/2 activation was prominent. PCNA mediates downstream cellular responses, including growth, survival, proliferation, migration, and angiogenesis, and the ERK1/2 pathway is essential for keratinocyte proliferation [22,23,24]. Increased pAKT activity further suggests that it promotes cell migration by enhancing the stability of cytoskeletal components or facilitating their remodeling. Studies have reported that the increased AKT expression promotes collagen formation and accelerates tissue regeneration in mouse wound models [25,26,27,28,29,30]. Collagen is important for maintaining skin elasticity, strength, and moisture, contributing to a firm and youthful appearance [31,32]. To investigate this further, collagen formation was analyzed in samples from an animal model [33]. H&E and collagen formation analyses of the treated animal models on day 4 showed that epithelial cell regeneration was evident and the number of fibroblasts in the dermal layer increased dramatically. Still, collagen formation had not yet occurred (Appendix A). However, in the tissues on day 10, the epithelial and dermal thicknesses increased to the level of normal skin. In addition, collagen expression and extracellular matrix (ECM) production, essential for skin regeneration, were significantly higher in the C24-LNP and EGF-positive treatment groups (Figure 5).

## 4. Discussion

This study highlights the potential of C24 ceramide lipid nanoparticles (C24-LNPs) as a novel and practical approach to enhancing skin wound healing. Our results demonstrate that hydrophobic C24 ceramide can be successfully converted into water-soluble nanoparticle formulations to improve bioavailability and therapeutic efficacy. The C24-LNP formulation exhibited uniform particle size and excellent solvent stability, which are important for enhanced cellular uptake and biological activity. In vitro, C24-LNP significantly promoted keratinocyte proliferation and migration, with effects comparable with or better than those of EGF protein. The upregulation of key proliferation markers such as Ki-67, PCNA, phosphorylated AKT, and ERK suggest that C24-LNP activates intracellular signaling pathways associated with cell growth and migration. In vivo experiments further validated the efficacy of C24-LNP in wound healing. The treated wounds showed accelerated healing, enhanced re-epithelialization, and increased dermal collagen compared with the control and EGF-treated groups. The histological results showed rapid and structural skin regeneration, confirming the ability of nanoparticles to promote functional and structural skin restoration. These results indicate that C24-LNP is an effective alternative to relatively expensive biological agents such as EGF.

Importantly, this study demonstrates that the composition of the nanoparticle, rather than its formulation, is important in achieving therapeutic effects. The lack of significant results with bare LNPs or isolated components highlights the importance of the nanoparticle composition design in delivering biologically active compounds to target tissues effectively. These findings pave the way for using ceramide-based nanoparticles in skin repair and regeneration therapies. By leveraging the properties of hydrophobic ceramide compounds and incorporating them as nanoparticle components, their functionality can be enhanced. Future studies should explore additional applications of C24-LNPs, including chronic wounds, burns, and skin diseases, and investigate their long-term safety, scalability, and clinical relevance.

## 5. Conclusions

This study establishes C24 ceramide lipid nanoparticles (C24-LNPs) as an advancement in skin wound-healing therapy. With the limited application potential of ceramide due to its hydrophobicity, the nanoparticle formulation achieves a water-soluble, stable, and bioavailable form of C24 ceramide. This development may be a practical proposal for exploiting the skin regeneration benefits of ceramide in clinical applications. Our results suggest that C24-LNP offers the dual benefits of biological efficacy and practical applicability. The fabricated nanoparticles enhance keratinocyte proliferation and migration by activating essential molecular biology pathways. These cellular responses, which have traditionally relied on biologics such as EGF protein, were achieved using C24-LNP alone, implying its potential to replace or complement existing growth factor-based therapies. From the results in animal models, the synthesized C24-LNP material demonstrates histological recovery of enhanced dermal and epidermal regeneration, emphasizing its role in achieving comprehensive tissue repair. What sets this study apart is the shift from hydrophobic ceramide compounds to hydrophilic via nanoparticulation as a key determinant of efficacy. These findings are expected to open new avenues for cost-effective and scalable treatments in dermatology beyond acute wound healing to chronic wounds, burns, and conditions involving compromised skin barriers.

## Figures and Tables

**Figure 1 pharmaceutics-17-00242-f001:**
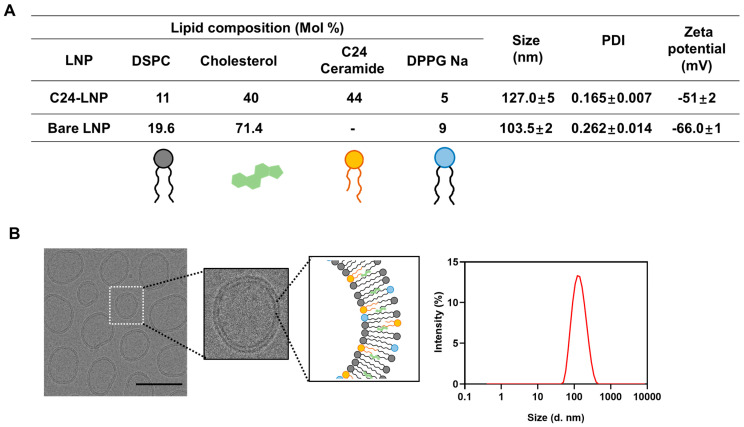
Composition and physical properties of C24-LNP. (**A**) Summary of components and physical analysis results. (**B**) Cryo-EM images and their enlarged images (scale bar: 100 nm). The size distribution results of C24-LNP synthesized with an optimized composition and ratio through DLS analysis. The synthesized C24-LNP showed solution stability for more than 3 months, confirmed through DLS measurements (Appendix A). The size, PDI, and zeta potential values are the results of measurements taken three times per batch from three synthesis batches.

**Figure 2 pharmaceutics-17-00242-f002:**
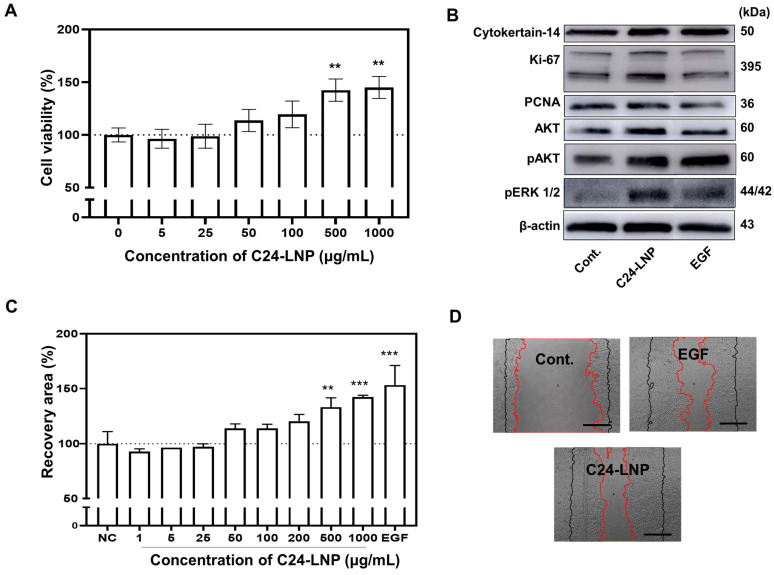
In vitro proliferation and migration assay for C24-LNP. (**A**) Cell viability data according to the treatment concentration of C24-LNP. Mean ± SEM, *n* = 5, ** *p* < 0.05 vs. control. (**B**) Expression of biomarkers related to the proliferation of cells treated with C24-LNP and EGF by Western blot analysis. (**C**) Confirmation of the migration effect of cells treated with C24-LNP and EGF. Raw data are shown in the Appendix A. (**D**) The picture shows a representative image (the red line indicates the case after cell growth, and the black line indicates the initial scratch area; scale bar: 500 µm). *** *p* < 0.005 vs. control, ** *p* < 0.05 vs. control.

**Figure 3 pharmaceutics-17-00242-f003:**
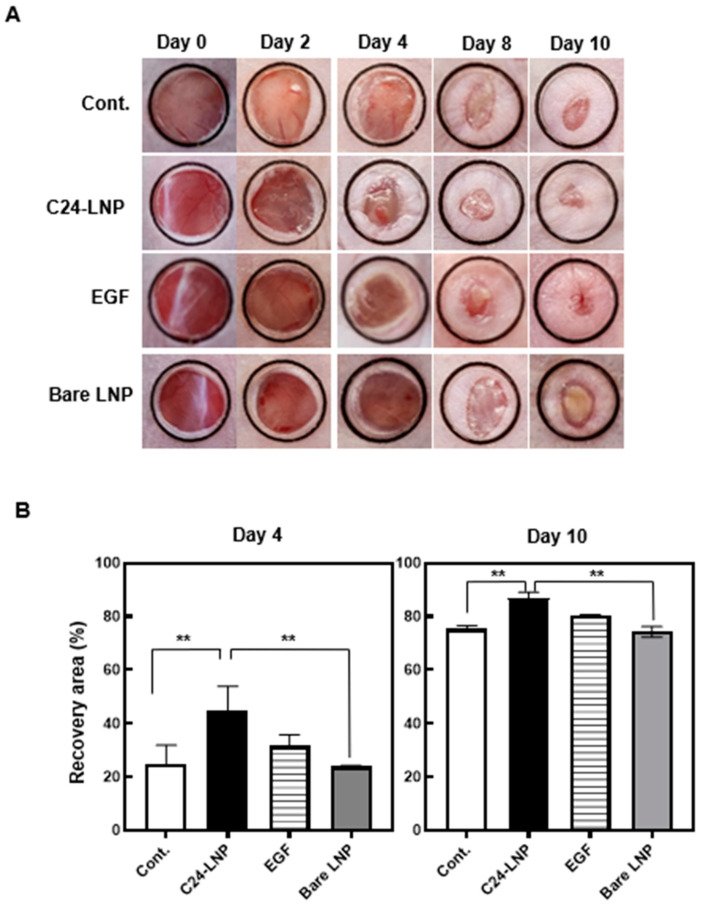
In vivo wound-healing assessment. (**A**) The animal models were prepared by making holes of uniform size using a 6 mm biopsy punch, and the recovery was observed daily for various samples. The black circles indicate identical observation areas. (**B**) The recovery area was measured three times and observed. Mean ± SEM, *n* = 3, ** *p* < 0.05 vs. control or bare LNP.

**Figure 4 pharmaceutics-17-00242-f004:**
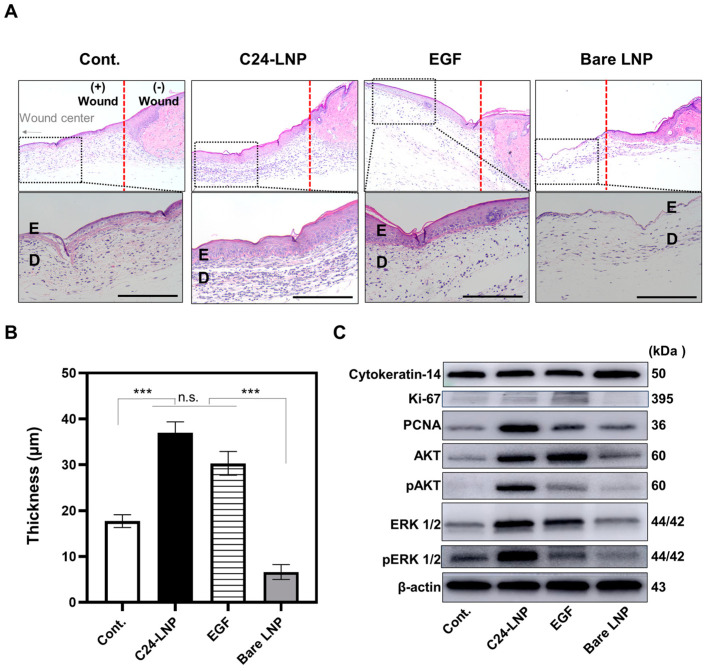
Histological analysis of an in vivo wound-healing animal model. (**A**) Microscopic images of each treated tissue after H&E staining (scale bar: 100 µm). The red dotted line indicates the boundary of the hole made using a punch, and the photos below are enlarged portions of those above (E and D indicate the epidermis and dermis, respectively). (**B**) The epidermis was significantly thicker in the positive control group, C24-LNP, and EGF treatment groups. The statistical difference was compared among each group. Mean ± SEM, *n* = 7, *** *p* < 0.005, n.s.: *p* > 0.05. (**C**) Analysis of proliferation-related biomarkers in the tissue. Strong expressions were observed in the positive control group.

**Figure 5 pharmaceutics-17-00242-f005:**
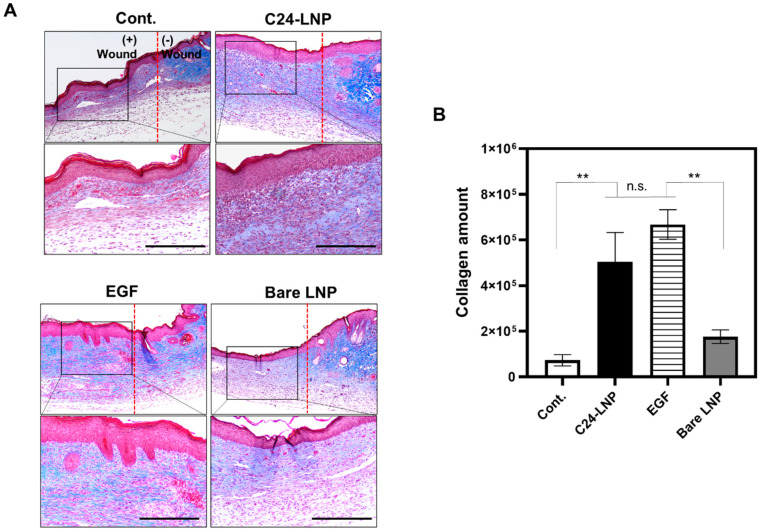
Analysis of collagen formation in tissues. (**A**) Results confirming the distribution of collagen protein in the tissues of the 10 days recovery group (blue is collagen staining). The red dotted line indicates the boundary where holes were formed using a punch. The picture below is an enlarged version of that above (scale bar: 100 µm). (**B**) A diagram expressing the intensity of collagen protein from the tissues. In the 4-day recovery group, collagen formation hardly progressed (Appendix A). The statistical difference was compared among each group. Mean ± SEM, *n* = 7, ** *p* < 0.05, n.s.: *p* > 0.05.

## Data Availability

Data are contained within the article and Appendix A.

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
