# Peer review of "C24 Ceramide Lipid Nanoparticles for Skin Wound Healing"

_pharmaceutics, 2025, doi:10.3390/pharmaceutics17020242_

Round 1

Reviewer 1 Report

Comments and Suggestions for Authors

The article is prepared carefully, only a few comments:

Line 80. There is an error in the name of the chemical compound. Also, it makes sense to use a more common name 1,2 dipalmitoyl-sn-glycero-3-phospho-(1'-rac-glycerolsodium salt.

It is advisable to clarify the captions of the Figures. The caption of Figure 3B does not match the figure itself, there are no black dotted lines or red lines.

Author Response

Reviewer 1

The article is prepared carefully, only a few comments:

Comments 1.

Line 80. There is an error in the name of the chemical compound. Also, it makes sense to use a more common name 1,2 dipalmitoyl-sn-glycero-3-phospho-(1’-rac-glycerol) sodium slat.

: Thank you for the precise point. We have corrected it with the name you mentioned.

Comments 2.

It is advisable to clarify the captions of the Figures. The caption of Figure 3B does not match the figure itself, there are no black dotted lines or red lines.

: Thank you very much for pointing out the error in the first draft of the paper. We have removed the captions and added an explanation in caption 3(A). 

Reviewer 2 Report

Comments and Suggestions for Authors

The current study aimed to fabricate ceramide-loaded lipid nanoparticles for wound healing, the study is interesting. However the following  should be addressed:

1-Several previous results have incorporated ceramides into microemulsions, nanoemulsions, liposomes, nanoparticles and microparticles. Please address the novelty of the current study with a summarization of the findings of previous studies in the introduction section.

2-Please provide a reference for the following dose 10 µL of C24- 142 LNP (1 mg/mL in DW) was topically applied daily for 10 days starting from the day of  injury (day 0), line 143.

3-Please provide the weight of mice utilized in the in vivo study.

4-Please provide SD values for PS, PDI, & ZP values.

5-Additional references are required in the result section.

6- Please discuss in detail the optimization of lipid nanoparticle components.

7-The authors did not mention the determination of the morphology of lipid nanoparticles in the experiment section; the Cryo-EM image is demonstrated in the result section.

Author Response

Reviewer 2

The current study aimed to fabricate ceramide-loaded lipid nanoparticles for wound healing, the study is interesting. However the following should be addressed:

Comments 1.

Several previous results have incorporated ceramides into microemulsions, nanoemulsions, liposomes, nanoparticles and microparticles. Please address the novelty of the current study with a summarization of the findings of previous studies in the introduction section.

: Thank you for your comments, which could improve the quality of the paper.

This study’s novelty is that C24 ceramide, which has a long alkyl chain with strong hydrophobic characteristics, was effectively manufactured as lipid nanoparticles. The composition of C24-LNP was established to maintain a physiochemically stable form and enable bio-application. No report has been made on the actual wound-healing effect of the C24 ceramide manufactured as lipid nanoparticles. This point is different from previous studies. A brief explanation of this point has been added to the Introduction section (lines 71 ~ 76).

Comments 2.

Please provide a reference for the following dose 10 uL of C24-LNP (1mg/mL in DW) was topically applied daily for 10 days starting from the day of injury (day 0), line 143.

: Thank you for allowing us to add references that are necessary to help readers understand. We have added the relevant paper as number 19.

Comments 3.

Please provide the weight of mice utilized in the in vivo study.

: Thanks for pointing out the part we missed. Added weight at Line 145.

Comments 4.

Please provide SD values for PS, PDI, and ZP values.

: We added SD values to the main text and Figure 1A.

Comments 5.

Additional references are required in the result section.

: We added a paper related to LNP preparation in reference number 20.

Comments 6.

Please discuss in detail the optimization of lipid nanoparticle components.

: Thank you for improving the quality of the paper by providing additional explanations. We have added additional discussion in the main text for optimizing the LNP composition (Line 186 ~ 191).

Comments 7.

The authors did not mention the determination of the morphology of lipid nanoparticles in the experiment section; the Cryo-EM image is demonstrated in the result section.

: Thank you for pointing out the shortcomings in our explanation. We added the Cryo-EM analysis description to the experimental section (Line 96 ~ 100).

Reviewer 3 Report

Comments and Suggestions for Authors

The article is generally well written and has an acceptable structure.

The article is light on nanoparticles formulation and process development. Further data is needed for a complete and comprehensive article. While formulation optimization does not need to be included, more analytical data in regard of C24-NLP and bare NLP is needed. Thus, a major review is advised.

I could not access some material – i.e. Figures starting with S – I guess this is a part of the supplement which was not present in the review material and thus cannot be assessed. Please provide this material in the new review.

Page 2, line 60: The reference does not state that C24 is »useful for repairing and strengthening damaged skin barriers«. Please revise and find a suitable reference.

Page 2, line 60: More detail is needed for the NP preparation. When the two phases were mixed, did you apply agitation? How? How much? The same goes for ethanol evaporation – temperature, time, mixing….

Page 2, line 60: How were bare LNP prepared? This should also be mentioned in the methods. Are there any analytical results for bare NLP available? This should also be provided.

Page 2, line 110: What was the concentration of C24-LNP?

Page 4, line 170: Is this optimization described somewhere? If yes, then please provide a reference, otherwise state that data is not shown.

Page 4, line 173: Please provide the number of measurements for each batch of nanoparticles and the number of batches measured. Additionally, please provide the variations (*/- SD) in size, PDI and zetapotential of NLPs. This is critical to confirm and understand the variability of your process. Further, are there any stability data available? If not, this should be listed as a limitation.

Page 5, line 194: »When bare LNPs…« the sentence is misleading, please rewrite. Further, please provide the data for the bare LNPs. There is no mentioning of concentration used and the numbers of experiment

Page 6, line 212: Concentration above 500 not 50

Comments on the Quality of English Language

At time sentences are poorly formulated and unclear. A review by a native speaker is advised.

Author Response

Reviewer 3

The article is generally well written and has an acceptable structure. The article is light on nanoparticles formulation and process development. Further data is needed for a complete and comprehensive article. While formulation optimization does not need to be included, more analytical data in regard of C24-LNP and bare LNP is needed. Thus, a major review is advised.

Comments 1.

I could not access some materials – i.e. Figures starting with S- I guess this is a part of the supplement which was not present in the review material and thus cannot be assessed. Please provide this material in the new review.

: We are sorry to hear that the supplementary materials were not delivered. We submitted it correctly, and it was probably missed during the process. It will be delivered in the system.

Comments 2.

Page 2, line 60: The reference does not state that C24 is “useful for repairing and strengthening damaged skin barrier” please revise and find a suitable reference.

: We agree with your point and appreciate it. We have confirmed that the reference you provided was inaccurate, so we have revised it (reference 10).

Comments 3.

Page 2, line 60: More detail is needed for the NP preparation. When the two phase were mixed, did you apply agitation? How? How much? The same goes for ethanol evaporation-temperature, time mixing….

: We agree that our explanation was not specific enough. We have revised it to include specific amounts and times, as mentioned (Line 91 ~ 93).

Comments 4.

Page 2, Line 60: How were bare LNP prepared? This should also be mentioned in the methods. Are there any analytical results for bare LNP available? This should be provided.

: We added the description of the preparation of bare LNP and added analysis data in Figure 1A.

Comments 5.

Page 2, Line 110: What was the concentration of C24-LNP

: We presented the concentration of C24-LNP when it was treated in the cells.

Comments 6.

Page 2, Line 170: Is this optimization described somewhere? If yes, then please provide a reference, otherwise state that data is not shown.

: We have added the necessary reference (number 20).

Comments 7.

Page 4, Line 173: Please provide the number of measurements for each batch of nanoparticles and the number of batches measured. Additionally, please provide the variations (*/- SD) in size, PDI and zetapotential of LNPs. This is critical to confirm and understand the variability of your process. Further, are there stability data available? If not, this should be listed as a limitation.

: We added the number of measurements per nanoparticle batch and the number of measured batches, along with the SD values to Figure 1. And the solution stability data were added to Figure S1.

Comments 8.

Page 5, Line 194: “When bare LNPs ~” the sentence is misleading, please rewrite. Further, please provide the data for the bare LNPs. There is no mentioning of concentration used and the numbers of experiments.

: Thank you for your point. We have corrected the sentence (Line 216) and added information about bare LNP to Figure 1A.

Comments 9.

Page 6, Line 212: Concentration above 500 not 50.

: Thank you for correcting our errors. We have corrected the values.

Reviewer 4 Report

Comments and Suggestions for Authors

The authors consider ceramіd-C24 as a wound care product that is more accessible than growth factors. Based on in vitro and in vivo experiments they convincingly prove that included in the liposome shell ceramide acquires the properties of water solubility, becomes bioavailable, and can cause accelerated wound healing at the level with the EGF growth factor, which is used as a positive comparison. These results are important for science in terms of identifying the role of ceramides in their external delivery in tissue regeneration processes. The developed methodology can also have practical application in the development of new effective wound care products. I believe that the article can be published with minimal amendments. 

Comments:

Overall, the readability and quality of English are good. However, there are some punctuation errors. Certain words and phrases should be replaced with more formal alternatives suitable for scientific writing, e.g. consider replacing “we realized” with “we observed” or “we suggest”

1.     Introduction

1.1.  In the introduction, it is worth expanding the information on the importance and possibility of creating ceramide delivery systems through the development of water-soluble forms capable of transporting them into the aqueous medium (inclusion in micelles, microemulsions and other delivery vehicles).

1.2.  There is also evidence that ceramides in the form of hydrophobic water-soluble materials can even be harmful (Schild J, Kalvodová A, Zbytovská J, Farwick M, Pyko C. The role of ceramides in skin barrier function and the importance of their correct formulation for skincare applications. Int J Cosmet Sci. 2024 Aug;46(4):526-543. doi: 10.1111/ics.12972. PMID: 39113291.) in tissue regeneration processes, so this fact is important to discuss in the introduction.

1.3.  The next sentence may be inappropriate: "lipid nanoparticles (LNPs), such as liposomes, will effectively provide hydrophilic properties to ceramide". Formally, due to the inclusion in the liposome, ceramide can be transferred to an aqueous medium, but this does not mean that it has become hydrophilic. Line 68-69.

2.     Experimental Methods

1.1  It is not very clear how much active substance (C24) was contained in the dose of 10 μl. 1 mg/ml is the concentration for liposomes with C24 or pure ceramide C24 in the dispersion of C24-LPN nanoparticles? Line 143

3. Results

1.1  The component composition was optimized to achieve higher stability in water, but it is not specified how long this stability lasts. Is it possible to control the size of nanoparticles? Line 170-171

1.2  Why in Figures 2a and 2c there are breaks on the Y axis, but at the beginning and end of the break there are the same values (50)? Line 221

1.3  The capture of Fig. 2 indicates "the picture on the right," but here it is better to designate the letter and mark the figure with a number and a letter like others. Line 226

1.4  How could you explain the fact that the advance in wound closing when using S24-LPN significantly decreased from the 4th day to the 10th day? Figure 3. Line 239

4. References

1.     The reference list does not correspond to the journal's guidelines

Author Response

Reviewer 4

The authors consider ceramide-C24 as a wound care product that is more accessible then growth factors. Based on in vitro and in vivo experiments they convincingly prove that included in the liposome shell ceramide acquires the properties of water solubility, becomes bioavailable, and can cause accelerated wound healing at the level with the EGF growth factor, which is used as a positive comparison. These results are important for science in terms of identifying the role of ceramides in their external delivery in tissue regeneration processes. The developed methodology can also have practical application in the development of new effective wound care products. I believe that the article can be published with minimal amendments.

Comments:

Overall, the readability and quality of English are good. However, there are some punctuation errors. Certain words and phrases should be replaced with more formal alternatives suitable for scientific writing, e.g. consider replacing “we realized” with “we observed” or “we suggest”

: Thank you for pointing out the overall readability of the English. We have made corrections.

  1. Introduction

1.1. In the introduction, it is worth expanding the information on the importance and possibility of creating ceramide delivery systems through the development of water-soluble forms capable of transporting them into the aqueous medium (inclusion in micelles, microemulsion and other delivery vesicles).

: Thanks for pointing out that we could add extensibility to our technology in the introduction. We added the explanation (Line 80 ~ 82).

1.2. There is also evidence that ceramides in the form of hydrophobic water-soluble materials can even be harmful (Schild J, Kalvodova A …. The role of ceramide in skin barrier function and the importance of their correct formulation for skincare applications. Int. J. Cosmet. Sci. 2024) in tissue regeneration processes, so this fact is important to discuss in the introduction.

: Thanks for adding great information. We added an explanation and reference (Line 76 ~ 78).

1.3. The next sentence may be inappropriate: “lipid nanoparticles (LNPs, such as liposome, will effectively provide hydrophilic properties to ceramide”. Formally, due to the inclusion in the liposome, ceramide can be transferred to an aqueous medium, but this does not mean that it has become hydrophilic. Line 68-69.

: We totally agree with your point and appreciate it. We have corrected it to make it more accurate.

  1. Experimental Methods

  • It is not very clear how much active substance (C24) was contained in the dose of 10uL. 1mg/mL is the concentration for liposomes with C24 or pure ceramide C24 in the dispersion of C24-LNP nanoparticles? Line 143.

: We added an accurate amount of C24 (Line 158).

  1. Results

1.1. The component composition was optimized to achieve higher stability in water, but it is not specified how long this stability lasts. Is it possible to control the size of nanoparticles? Line 170-171

: We measured the stability of C24-LNP in aqueous solution by DLS and added the results in Figure S1.

1.2. Why in Figures 2a and 2c there are breaks on the Y axis, but at the beginning and end of the break there are the same values (50)? Line 221

: I broke below 50. I modified the y-axis. Thank you.

1.3. The capture of Fig. 2 indicates “the picture on the right”, but here it is better to designate the letter and mark the figure with a number and a letter like others. Line 226

: Thanks for the detailed comment. We corrected it.

1.4. How could you explain the fact that the advance in wound closing when using C24-LNP significantly decreased from the 4th day to the 10th day? Figure 3. Line 239.

: This is thought to be observed because the dermal layer growth process is dominant until the fourth day, when the epidermal layer grows rapidly thereafter. This explanation has been added to the text (Line 262 ~ 264).

  1. References
  2. The reference list does not correspond to the journal’s guidelines

: We have double-checked the reference format. Thank you.

Round 2

Reviewer 2 Report

Comments and Suggestions for Authors

The authors have addressed all the requested comments.

Author Response

Reviewer 2 had no additional comments.

Reviewer 3 Report

Comments and Suggestions for Authors

Reference 10: the reference was changed, but it still does not support the claim stated in your article. You are claiming that »In addition, C24 is particularly useful for repairing and strengthening damaged skin barriers« The new reference claims no such thing. It does not even mention any ceramide as having the above claimed properties or their function in the skin. Ceramides are proposed as a prediction factor for metabolic syndrome in the cited reference.

Comments 6.

Page 2, Line 170: Is this optimization described somewhere? If yes, then please provide a reference, otherwise state that data is not shown.

: We have added the necessary reference (number 20).

The reference only describes general principles for liposome formation and preparation. It does not describe your formulation. You are claiming that “We precisely optimized the neutral lipid, cholesterol, and anionic lipid compound ratio to prepare”. This is a strong claim that is not supported. However, you explain that you tried to keep cholesterol amount as high as possible and that 5% of anionic lipid is needed – probably to ensure aqueous solubility and stability. This is not “precise optimization” therefore I would omit such strong wording and keep to the high-level overview that you provided.

Figure S2:

There is no S2C although it is referenced in Figure S2 description.

Figure S2C is not referenced in the article.

Why are the ratios of C24-NLP and EGF treated cells different in Figure S2B and S2A? Maybe due to different sampling times – this is unclear from the text.

Comments on the Quality of English Language

The article is substantially improved, well done.

By far the biggest issue is the language. The English is not up to the expected standard. The sentences are sometimes awkward. (e.g. “we have been attempting to maintain the cholesterol content as much as possible to maintain structural stability” or “We have confirmed that a minimum of 5 % of anionic lipid compounds, such as DPPG, must have aqueous solution stability”). A review by a native speaker versed in scientific writing is strongly advised.

Author Response

Reviewer 3
Reference 10: the reference was changed, but it still does not support the claim stated in your article. You are claiming that »In addition, C24 is particularly useful for repairing and strengthening damaged skin barriers« The new reference claims no such thing. It does not even mention any ceramide as having the above claimed properties or their function in the skin. Ceramides are proposed as a prediction factor for metabolic syndrome in the cited reference.

:Thank you for allowing me to add a more precise study to the reference. I have changed reference 10. This previous study confirmed that when C16 ceramide increases and C24 ceramide decreases, the lipid composition of the skin barrier changes, which increases skin permeability. C24 ceramide plays an important role in strengthening the skin barrier, and its decrease is related to a decline in skin barrier function. I have changed the study to reference 10.

Comments 6.
Page 2, Line 170: Is this optimization described somewhere? If yes, then please provide a reference, otherwise state that data is not shown.
: We have added the necessary reference (number 20).
The reference only describes general principles for liposome formation and preparation. It does not describe your formulation. You are claiming that “We precisely optimized the neutral lipid, cholesterol, and anionic lipid compound ratio to prepare”. This is a strong claim that is not supported. However, you explain that you tried to keep cholesterol amount as high as possible and that 5% of anionic lipid is needed – probably to ensure aqueous solubility and stability. This is not “precise optimization” therefore I would omit such strong wording and keep to the high-level overview that you provided.

: 5page Lin3 191, corrected to the correct wording.

Figure S2:
There is no S2C although it is referenced in Figure S2 description.
Figure S2C is not referenced in the article.

:Thank you for your accurate point. I had mistakenly placed Figure 4C in the caption of reference 2C while writing the paper. I have deleted the description of supplementary Figure S2C.

Why are the ratios of C24-NLP and EGF treated cells different in Figure S2B and S2A? Maybe due to different sampling times – this is unclear from the text.

:The difference in the ratio between S2A and S2B is that S2A is expressed in vitro, while S2B is expressed in in vivo samples. However, the commonality in expression rates is that expression was mostly induced higher in the C24-LNP treatment group and the EGF treatment group compared to the control group. Thank you for pointing out that we did not express this accurately. We added an explanation in the supplementary caption (Fig. S2).

The article is substantially improved, well done.

By far the biggest issue is the language. The English is not up to the expected standard. The sentences are sometimes awkward. (e.g. “we have been attempting to maintain the cholesterol content as much as possible to maintain structural stability” or “We have confirmed that a minimum of 5 % of anionic lipid compounds, such as DPPG, must have aqueous solution stability”). A review by a native speaker versed in scientific writing is strongly advised.

: We agree with the criticism and have completed the English proofreading through the MDPI system.